# Application of modular response analysis to medium- to large-size biological systems

**Meriem Mekedem** [1,2,3], **Patrice Ravel** [1,2,3,4], **Jacques Colinge** [1,2,3,5] *

1 Université de Montpellier, Montpellier, France, 2 Institut de Recherche en Cancérologie de Montpellier, Inserm U1194, Montpellier, France, 3 Institut régional du Cancer Montpellier, Montpellier, France, 4 Faculté de Pharmacie, Université de Montpellier, Montpellier, France, 5 Faculté de Médecine, Université de Montpellier, Montpellier, France

* jacques.colinge@inserm.fr

**Data Availability Statement:** All relevant data are within the manuscript and its Supporting Information files.

**Funding:** MM was supported by a PhD fellowship of the Algerian government. The funders had no

## Abstract

The development of high-throughput genomic technologies associated with recent genetic perturbation techniques such as short hairpin RNA (shRNA), gene trapping, or gene editing (CRISPR/Cas9) has made it possible to obtain large perturbation data sets. These data sets are invaluable sources of information regarding the function of genes, and they offer unique opportunities to reverse engineer gene regulatory networks in specific cell types. Modular response analysis (MRA) is a well-accepted mathematical modeling method that is precisely aimed at such network inference tasks, but its use has been limited to rather small biological systems so far. In this study, we show that MRA can be employed on large systems with almost 1,000 network components. In particular, we show that MRA performance surpasses general-purpose mutual information-based algorithms. Part of these competitive results was obtained by the application of a novel heuristic that pruned MRA-inferred interactions *a posteriori*. We also exploited a block structure in MRA linear algebra to parallelize large system resolutions.

## Author summary

The knowledge of gene and protein regulatory networks in specific cell types, including pathologic cells, is an important endeavor in the post-genomic era. A particular type of data obtained through the systematic perturbation of the actors of such networks enables the reconstruction of the latter and is becoming available at a large scale (networks comprised of almost 1,000 genes). In this work, we benchmark the performance of a classical methodology for such data called modular response analysis, which has been so far applied to networks of modest sizes. We also propose improvements to increase performance and to accelerate computations on large problems.

## Introduction

The expression and activity of genes and proteins in cells are controlled by highly complex regulatory networks involving genes and proteins themselves, but also non-coding RNAs, metabolites, etc. Despite tremendous efforts in research, including all the developments of high-

role in study design, data collection and analysis, decision to publish, or preparation of the manuscript.

**Competing interests:** The authors have declared that no competing interests exist.

throughput genomic technologies, a significant portion of this machinery remains uncharted. Dysregulations in such networks are related to many diseases. Moreover, healthy cells of a same organism often feature adjusted regulatory networks depending on their types and states. Techniques, both experimental and computational, that enable the inference of regulatory networks for different cells are obviously of great interest.

Reference databases such as Reactome [1], KEGG [2], IntAct [3], or STRING [4] compiling our knowledge of biological pathways or protein interactions have been established and provide valuable reference maps. Due to their universal nature, these maps do not reflect natural and pathologic variations of regulatory networks though some chosen disease pathways might be included [5,6]. In principle, researchers should generate data specific to the biological system of interest to assess the actual wiring of its regulatory network. Specific data can be combined with reference databases in some algorithms, while others only rely on *de novo* inferences. The field of systems biology has proposed many algorithms for such a purpose involving different approaches [7–9]. Obviously, algorithms must match the type of data available to perform their inferences such as a transcriptomes or proteomes obtained under multiple conditions, time series, or perturbation data.

In this work, we are interested in the inference of regulatory networks based on systematic perturbation data. That is, given a biological system of interest, which could be the whole cell, but also a small set of related genes or proteins such as a pathway or part of a pathway, we have access to information reporting the activity level of every component (gene/protein). Typical examples are transcript, protein, or phosphorylated protein abundances. This information is available in basal condition as well as under the systematic perturbation of each single component. When this type of data are obtained from a biological system in a steady state, modular response analysis [10] (MRA) has been widely and successfully applied [11]. The elegance of MRA is that it provides an efficient mathematical framework to estimate a directed and weighted network representing the system regulatory network. Most applications of MRA are limited to networks comprised of a modest number of modules (<10). In this study, we want to explore the application of MRA to medium- (>50) and large-size (>500) systems. It entails a particular implementation of the linear algebra at the heart of MRA to parallelize computations as well as the introduction of a heuristic to prune the inferred networks *a posteriori* to improve accuracy.

As stated above, rewiring of regulatory networks is natural and necessary to yield a multitude of cell types in higher organisms, and to adapt to distinct environmental conditions. Rewiring is also associated with several diseases [12,13], an extreme case being cancer [14–16]. For instance, kinase signaling cascades might be redirected in certain tumors to achieve drug resistance or to foster exaggerated cell growth. MRA has been applied to a number of such cancer-related investigations [17,18] considering rather small networks. Here, we take advantage of two published data sets that involve cancer cell lines and provide systematic perturbation data compatible with MRA requirements. The first–medium-size–data set [19] reports the transcriptional expression of 55 kinases and 6 non kinases under 11 experimental conditions (unstimulated plus 10 distinct stimulations). Under every condition, the transcript levels of all the 61 genes were obtained by shallow RNA sequencing, including wild type cells and cells with individual KOs of each gene. These data hence enable us to infer one network *per* condition (11 networks) to discover how those 61 genes regulate themselves transcriptionally. The second–large-size–data set was generated by the next generation of the Connectivity Map (CMap) using its new L1000 platform [20]. Both shRNA- and CRISPR/Cas9-based systematic perturbations of roughly 1,000, respectively 350, genes in 9, respectively 5, cell lines were released. These data enable us to infer 9+5 = 14 networks. To complement performance estimations based on real data sets, where complete and exact knowledge of the interaction is not

available, we also generated medium to large, realistic synthetic networks [21] and corresponding perturbation data.

We compare the performance of MRA, with and without the proposed pruning heuristic, to mutual information (MI)-based methods that have found broad acceptance. The adapted MRA implementation with optional heuristic post-processing is made available as an R script.

## Results

### Network inference algorithms

The availability of large functional genomics data collections (transcriptomes and/or proteomes) has led to the development of a number of algorithms aimed at inferring interaction networks [9]. An essential ingredient of most algorithms is the co-expression of genes (or proteins)[22], which can be captured by simple correlation coefficients [23], mutual information (MI), or diverse statistical models [24]. There are too many such algorithms to review them all here, but MI-based approaches seem to have provided off-the-shelf, robust solutions that are widely used. We hence compare MRA to representatives of this category such as CLR [25], MRNET [26], and ARACNE [27].

MI is often preferred over Pearson correlation for its ability to detect nonlinear relationships. With a network involving $n$ genes whose expression levels are measured in $m$ transcriptomes, we write $X_i$ the discrete distribution representing gene $i$ expression. The MI between genes $i$ and $j$ is given by

$$MI_{i,j} = H(X_i) + H(X_j) - H(X_i, X_j),$$

where $H(X) = -\sum_{k \in X} p(x_k) \ln (p(x_k))$ is the entropy of a discrete random variable $X$. There exist different estimators for $H(X)$ that use the $m$ available transcriptomes [28]. Networks of interactions identified though MI, imposing a minimal threshold on MI values, are commonly called relevance networks [29,30]. The CLR algorithm improves over relevance networks by introducing a row- and column-wise z-score-like transformation of $MI_{i,j}$ to normalize the MI matrix into a $Z = (z_{i,j})$ matrix before thresholding. Namely, for each gene $i$ CLR computes

$$z_i = \max \left\{ 0; \frac{MI_{i,j} - \text{mean}(MI_{i,.})}{\text{sd}(MI_{i,.})} \right\}$$

and then

$$z_{i,j} = \sqrt{z_i^2 + z_j^2}.$$

MRNET applies a greedy maximum relevance strategy to link each gene $i$ to the gene $j$ that has maximum MI with it ($j = \arg \max MI_{i,j}$). Additional links are added recursively maximizing MI with both the gene $i$ and the already linked genes until a stop criterion based on redundancy is met. A further approach by pruning was proposed by ARACNE authors, where as in relevance networks a common threshold is applied to all the $M_{i,j}$ followed by the application of a pruning rule. This rule states that, if gene $i$ interacts with gene $j$ through gene $k$, then $M_{i,j} \leq \min\{M_{i,k}; M_{k,j}\}$. Consequently, among each triplet of nonzero MI after initial thresholding, the weakest interaction is removed.

### The MRA and MRA+CLR algorithms

Due to its ability to model biological systems at various resolutions, the MRA terminology for a system component is a module. We follow this terminology and consider that the $n$ modules

composing the system have their activity levels denoted by $x \in \mathbb{R}^n$. Here, modules are genes and $x_i$ stands for gene $i$ transcript abundance. If we make the rather nonrestrictive assumption that relationships between modules are modeled by a dynamical system

$$\dot{x} = f(x)$$

($f(.)$ must exist but it does need to be known), and the system is in a steady state at the time of experimental measurements ($\dot{x} = 0$), MRA theory lets us compute an $n \times n$ matrix of local interaction strengths $r = (r_{i,j})$ from a gene $j$ to a gene $i$ $\left( r_{i,j} = \frac{\partial x_i}{\partial x_j} \frac{x_j}{x_i} \right)$. The matrix $r$ is obtained from linear algebraic computations based on the observed activity of each module in an unperturbed state, and under the individual, successive perturbations of each module. Details are provided in MRA original publication [10], reviews of MRA developments [11], or in our recent publication [17]. We use the notations of this recent paper. In Materials and Methods, we provide a brief overview of MRA along with a description of the particular way we implemented the linear algebra to take advantage of parallel computing.

Returning to the regulatory network inference problem, the MRA local interaction matrix $r$ provides us with a direct estimate of this network. Interactions are signed with positive coefficients representing activation and negative coefficients representing inhibition. Given the fact that we want to apply MRA to large systems, where every module does not necessarily have a direct influence on all the others, we also face the problem of thresholding or pruning. Within the context of this study, we call MRA the direct use of MRA computations followed by a threshold on the absolute values of $r$ coefficients (values below a given threshold in absolute values are set to 0). We also adapted CLR heuristic (z-score-like computation) to bring $r$ coefficients to a more uniform scale before thresholding. We call this algorithm MRA+CLR, see Materials and Methods for details.

## Application to synthetic data sets

In order to have access to an exact reference when testing inference algorithms, it is customary to generate realistic synthetic data. To start evaluating MRA and MRA+CLR, we took advantage of a recent network data generator called FRANK [21]. FRANK has the ability to generate regulatory networks that reach a steady-state and to output simulated expression data including individual perturbations, which obviously match our needs to test our application of MRA. FRANK-generated networks are defined by specifying the numbers of transcription factors (TFs) and target genes (TAs). TAs are genes that do not influence the expression of the other genes, while TFs do. Biologically speaking, FRANK's TFs do not need to be actual transcription factors, but they should rather be regarded as genes that can control the expression of other genes (including other TFs in FRANK). TFs hence include target genes that loop back into the regulatory network. We considered synthetic networks of different sizes and TF/TA ratios: 50TFx50TA, 100TFx100TA, 500TFx500TA, 1000TFx1000TA, and 75TFx1000TA.

We applied MRA, MRA+CLR, CLR, MRNET, and ARACNE to these networks. CLR and MRNET implementations were provided by the minet BioConductor package [28]. ARACNE implementation was provided by the parmigene BioConductor package [31]. To estimate performance, we compared inferences with the reference matrix used to generate each synthetic network data. To apply a uniform selection mechanism to all of the algorithms, we simply took the top 5%, 10%, 20%, 30% and 40% scores of the inferred interaction matrices. In some cases, ARACNE and MRNET returned less nonzero interactions than the number corresponding to top x%, in which case those algorithms were ignored at such a top x%. Representative performance is reported in Fig 1, and all the confusion matrices reporting true/false positives (TPs/FPs) and true/false negatives (TNs/FNs) along with specificity, accuracy, precision, recall,

and a P-value for the significance of the intersection with the reference network (hypergeometric test) are provided in S1 Table. Because all the inferred interactions were selected in identical numbers for each algorithm, all the numbers in the confusion matrices as well as specificity, precision, etc. are coupled. We thus only report TPs to represent the relative performances of the algorithms in Fig 1B and 1C for the top 10% and 20% selection levels.

The three MI-based algorithms (CLR, MRNET, and ARACNE) as well as MRA performed similarly. MRA+CLR consistently achieved superior performance.

## Application to a medium-size data set

Gapp *et al.*[19] published a data set, where they studied the transcriptional impact of the full knockouts (KOs) of 55 tyrosine kinases and 6 non-kinases. We call this data set K61. The systematic perturbations (KOs) of each gene as well as the unperturbed transcriptomes obviously constitute a *bona fide* MRA data set. The transcriptomes were acquired under 11 conditions: no stimulation (None), FGF1, ACTA, BMP2, IFNb, IFNg, WNT3A, ionomycin (IONM), resveratrol (RESV), rotenone (ROTN), and deferoxamine (DFOM) stimulation. Stimulations were applied for 6 hours allowing the cells to adapt and reach a steady state or near steady state. To facilitate the generation of full-KOs, human HAP1 haploid cells [32] were utilized. The published transcriptomes were not limited to the expression of the 61 perturbed genes, but here, due to the specifics of MRA, we limited the data to those 61 genes. Replicates were essentially averaged (see Materials and Methods), resulting in a 61×61 matrix for each of the 11 conditions

We applied MRA, MRA+CLR, CLR, MRNET, and ARACNE to each of the 11 conditions in the K61 data set separately. To estimate performance, we compared our results with the STRING database [4] due to its broad content. Indeed, working with transcriptomic data implies that the inferred networks might overlap protein complexes as well as certain parts of known pathways, but they might also unravel different types of relationships such as genetic interactions, strong co-regulation, etc. Physical interactions only of well-described pathway databases [1,3] might thus be too restrictive, hence the choice of STRING. As above, we took the top 5%, 10%, 20%, 30% and 40% scores of the inferred interaction matrices by each algorithm and determined the intersection with STRING. Since STRING interactions are provided with a confidence score, we used STRING score > 0.5 interactions as a default. Intersections based on STRING scores > 0 and > 0.8 are provided in S2 Table. Intersection with STRING resulted in confusion matrices and derived indicators (P-value, sensitivity, etc.) similar to what we obtained for the synthetic data. A representative example (None condition) is featured in Fig 2A, while complete results are in S2 Table.

Considering that the STRING database is both noisy and incomplete, and it only reflects a universal interactome, confusion matrices derived from STRING (or any such database) are rough approximations. Nonetheless, the use of a constant reference and identical selection criteria for all the algorithms make the observed relative performances a reliable indicator of actual differences. Due to our top x% selection mechanism and the use of a constant reference, all the numbers in confusion matrices as well as P-values, recall, etc. are coupled. Accordingly, we only report TPs for the 11 conditions of the data set at the top 10% and the top 20% selection levels in Fig 2B to 2E. Similar to the synthetic data sets, CLR, MRNET and ARACNE delivered comparable performance that was inferior to MRA+CLR. The advantage of MRA +CLR over MRA alone was not significant on K61 data. Results referring to more stringent (STRING score > 0.8) or less stringent (STRING score > 0) use of STRING interactions were very qualitatively close (S2 Table). We conclude that on these data as well, the MRA+CLR algorithm provides a competitive approach compared to MI-based solutions.

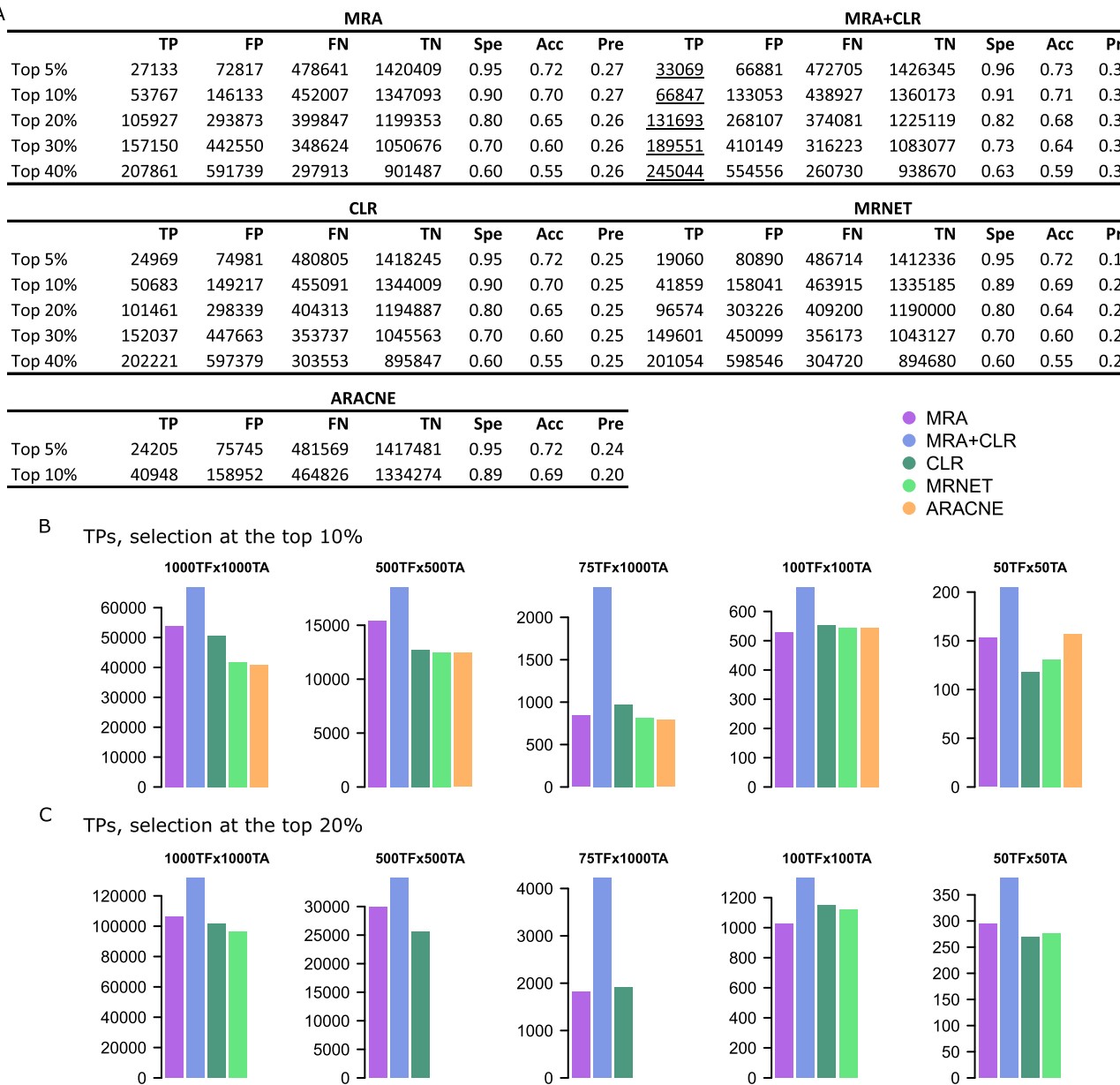

**Fig 1. Performance on synthetic data.** (A) Representative confusion matrices on the 1000TFx1000TA network. Spe = specificity, Acc = accuracy, and Pre = precision. Underlined values represent maxima. (B) TP numbers at the top 10% selection level in each network model. (C) TP numbers at the top 20% selection level. Note that in some cases, MRNET and ARACNE did not returned enough interactions to perform a number of selections equal to the top 20% of all possible interactions. We ignored those cases, hence the missing bars and numbers.

In their article, K61 authors discussed interesting differences in JAK1 *versus* JAK2 and TYK2 signaling, three members of the JAK family. In particular, they found that JAK1 KO cells were insensitive to IFNb and IFNg stimulation, while JAK2 and TYR2 KO cells responded normally although, in general, all these proteins are known to contribute to transcriptional response upon type I and II interferon stimuli [33]. To illustrate how network inference might provide some clue on such differences, we report in Fig 3A the MRA+CLR-inferred transcriptional interaction strengths between those three genes and their targets under the unstimulated (None), IFNb, and IFNg conditions. In the absence of stimulation, we clearly notice opposed

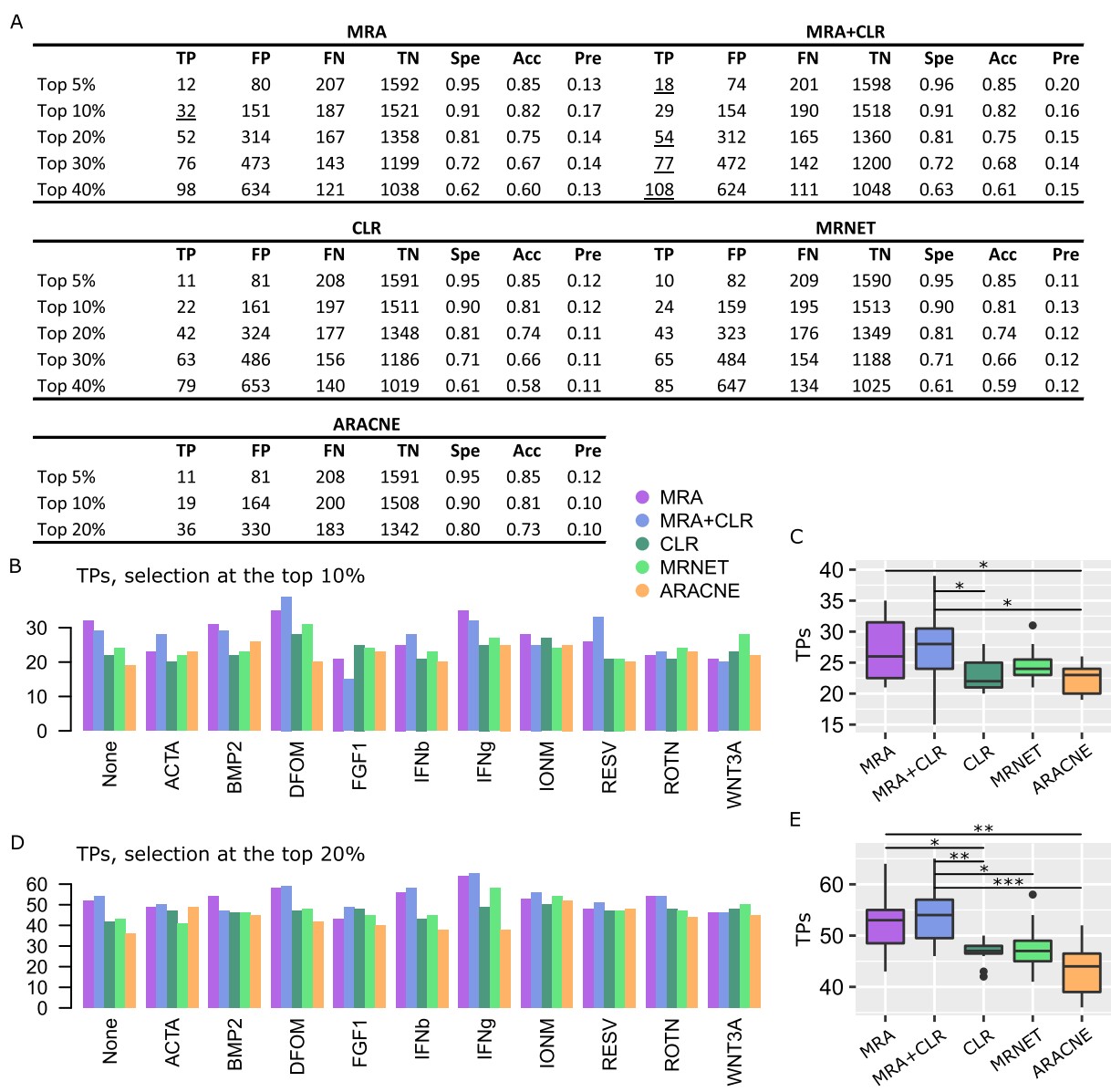

**Fig 2. Performance on K61 data against STRING interactions (STRING score > 0.5).** (A) Representative confusion matrices for the None condition. Spe = specificity, Acc = accuracy, and Pre = precision. Underlined values represent maxima. Note that in some cases, ARACNE did not return enough interactions to perform a number of selections equal to the top 30% or 40% of all possible interactions. We ignored those cases, hence the missing numbers. (B) TP numbers at the top 10% selection level. (C) Comparison between the algorithm TP numbers (Wilcoxon test, 2-sided, *P < 0.05). (D) TP numbers at the top 20% selection level. (E) Comparison between the algorithm TP numbers (Wilcoxon test, 2-sided, *P < 0.05, **P<0.005, ***P<0.001).

influences of JAK1 on its targets compared to JAK2 and TYR2 (first three columns), which already indicate different signal transduction capabilities. Upon IFNb stimulation, the interactions are closer with opposed action on ROR1 and PDFGRA. JAK2 and TYR2 remained highly similar in this condition. IFNg stimulation induced three different patterns with ROR1 transcriptional inhibition remaining a specific mark of JAK1. Gapp *et al.* also found differences in FGF receptors. FGF-induced response was attenuated in FGFR1 and FGFR3 KO cells, but preserved in FGFR2 and FGFR4 KO cells. In Fig 3B, we notice an almost perfect inversion of the activation/inhibition pattern between FGFR1 *versus* FGFR2 and FGFR3. FGFR4 adopted a

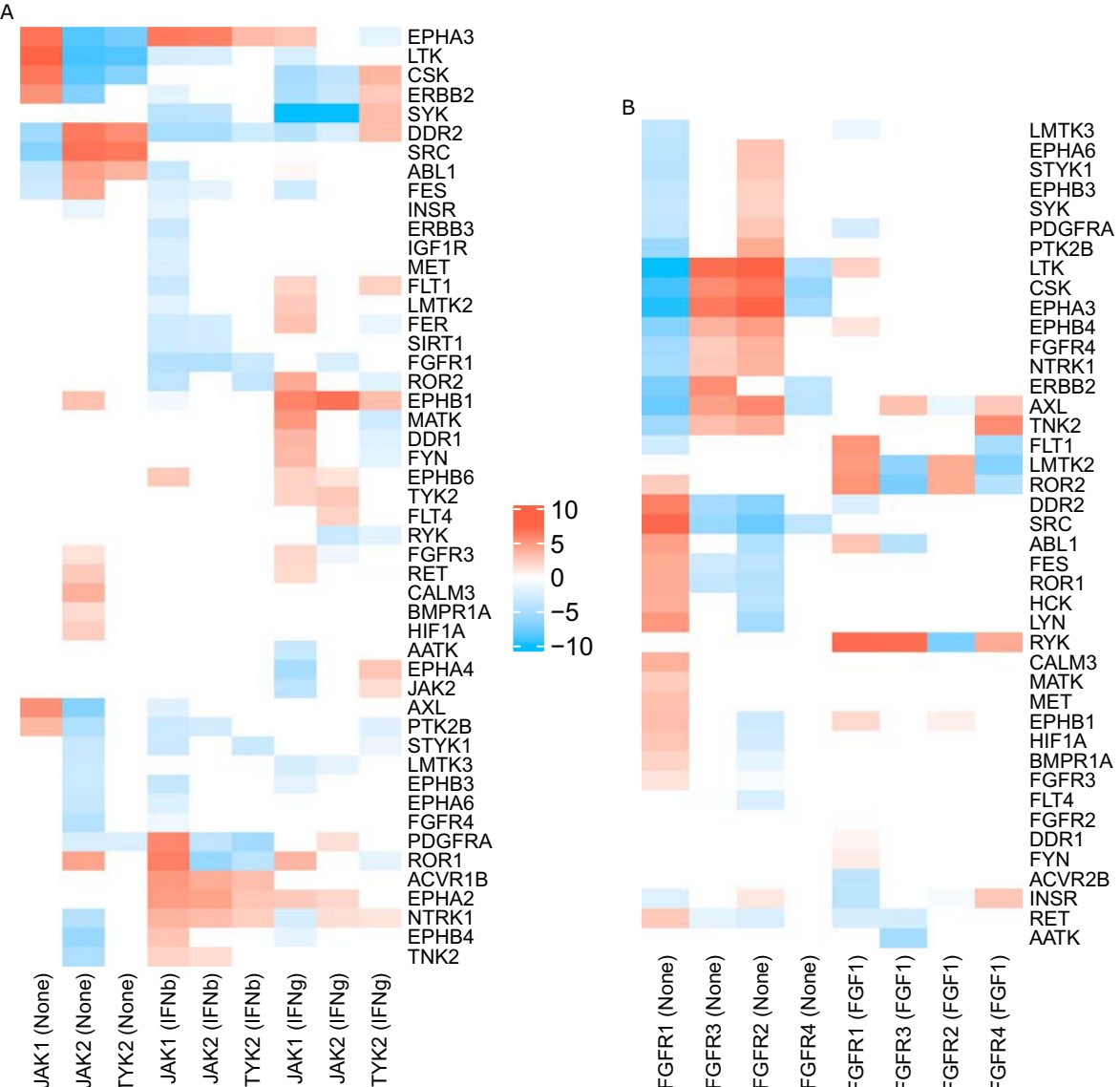

**Fig 3. MRA+CLR-inferred interactions (top 20% selected).** (A) Interaction strengths (in log₂ with sign preserved) between JAK1, JAK2, and TYR2 and their targets. Stimulatory conditions are in brackets (None, IFNb, IFNg). (B) Interaction strengths between FGFR1, FGFR2, FGFR3, and FGFR4 and their targets.

very different configuration with limited interactions. This observation already indicates a distinct role for FGFR1. Upon FGF stimulation, the interactions are patchier, but certain oppositions can be found such as a strong inhibitory action of FGFR1 and FGFR3 on RYK transcription.

## Application to a large-size data set

CMap next generation platform L1000[20] has recently released (December 2020) a new batch of data. These data are in majority comprised of transcriptomes obtained in reference cancer cell lines under a large number of perturbations with chemical agents, but most importantly shRNA-induced knockdowns and CRISPR/Cas9 KOs. L1000 cost effective design entailed the

identification of roughly 1,000 *hallmark* genes from which a large proportion of the whole transcriptome can be inferred. The L1000 platform only measures the expression of the hallmark genes experimentally. Two subsets of these data interest us. A first data set is composed of the almost systematic shRNA perturbation of all the hallmark genes, thus providing an expression matrix close to 1,000×1,000 in size for 9 human cell lines: A375 (metastatic melanoma), A549 (lung adenocarcinoma), HCC515 (non-small cell lung cancer, adenocarcinoma), HT29 (colorectal adenocarcinoma), HEPG2 (hepatocellular carcinoma), MCF7 (breast adenocarcinoma), PC3 (metastatic prostate adenocarcinoma), VCAP (metastatic prostate cancer), and HA1E (normal kidney cells). To alleviate shRNA off-target effects, L1000 employed multiple hairpins, which were integrated into a consensus gene signature (CSG) that the authors showed to be essentially devoid of off-target consequences [20]. Cells were harvested 96 hours after shRNA perturbation leaving time to reach a steady state that is compatible to shRNA common use. Due to variation in data production, the actual matrix sizes ranged from 815×815 (MCF7) to 938×938 (A375).

We followed the same performance evaluation procedure as above for K61. A representative (A375 cells) confusion matrix is reported in Fig 4A, followed by TP numbers at the top 10% and top 20% selection levels in Fig 4B to 4E. All the confusion matrices as well as results for different STRING score thresholds are in S3 Table. With L1000 shRNA larger matrices, but also knockdown perturbations instead of KOs, MRA and MRA+CLR advantage was much augmented over CLR and MRNET. The MRA+CLR algorithm outperformed MRA. ARACNE which performed moderately on K61 and synthetic data, achieved median performance almost identical to MRA+CLR with less variability. In every case, despite rather consistent differences between TPs predicted by the various algorithms, the large number of genes involved and potentially inaccuracies in the reference network STRING tend to make those differences small. This is reflected in the accuracy, precision, and specificity values that display essentially no variation.

To illustrate the interest of network inference at this scale, we intersected MRA+CLR inferences in normal kidney HA1E and melanoma A375 cells with a Gene Ontology term, *i.e.*, GO:0006974 cellular response to DNA damage stimulus. In Fig 5, we can notice the difference in connectivity between normal cells and cells where this process is obviously exacerbated, in particular the regulation of ATMIN a key molecule in DNA repair. This result is in agreement with the known rewiring of genetic networks in response to DNA damage [34].

The second L1000 data set of interest is the CRISPR/Cas9 collection of KOs. These data were only available for five cell lines: A375, A549, HT29, MCF7, and PC3. The matrix sixes ranged from 343×343 (MCF7) to 359×359 (A375). Performance results are featured in Fig 6 and S4 Table. Although MRA and MRA+CLR again dominated the other algorithms, their advantage was less pronounced on these large, full KO data. ARACNE median performance was inferior to MRA+CLR with similar or higher variability. The observation we made above regarding the size of the biological system squeezing differences in performance remains valid here, although the problem is slightly attenuated with the smaller CRISPR/Cas9 data sets compared to their shRNA equivalent.

## Execution times

We compared execution times of each algorithm on a server equipped with Intel Xeon E7-4870 processors running at 2.4 Ghz, Fig 7A. MI-based algorithms were extremely fast while MRA-based algorithms required much more compute time. The parallelized implementation of MRA algebra that we propose here (see Materials and Methods) allowed us to substantially reduce MRA+CLR compute times, Fig 7.

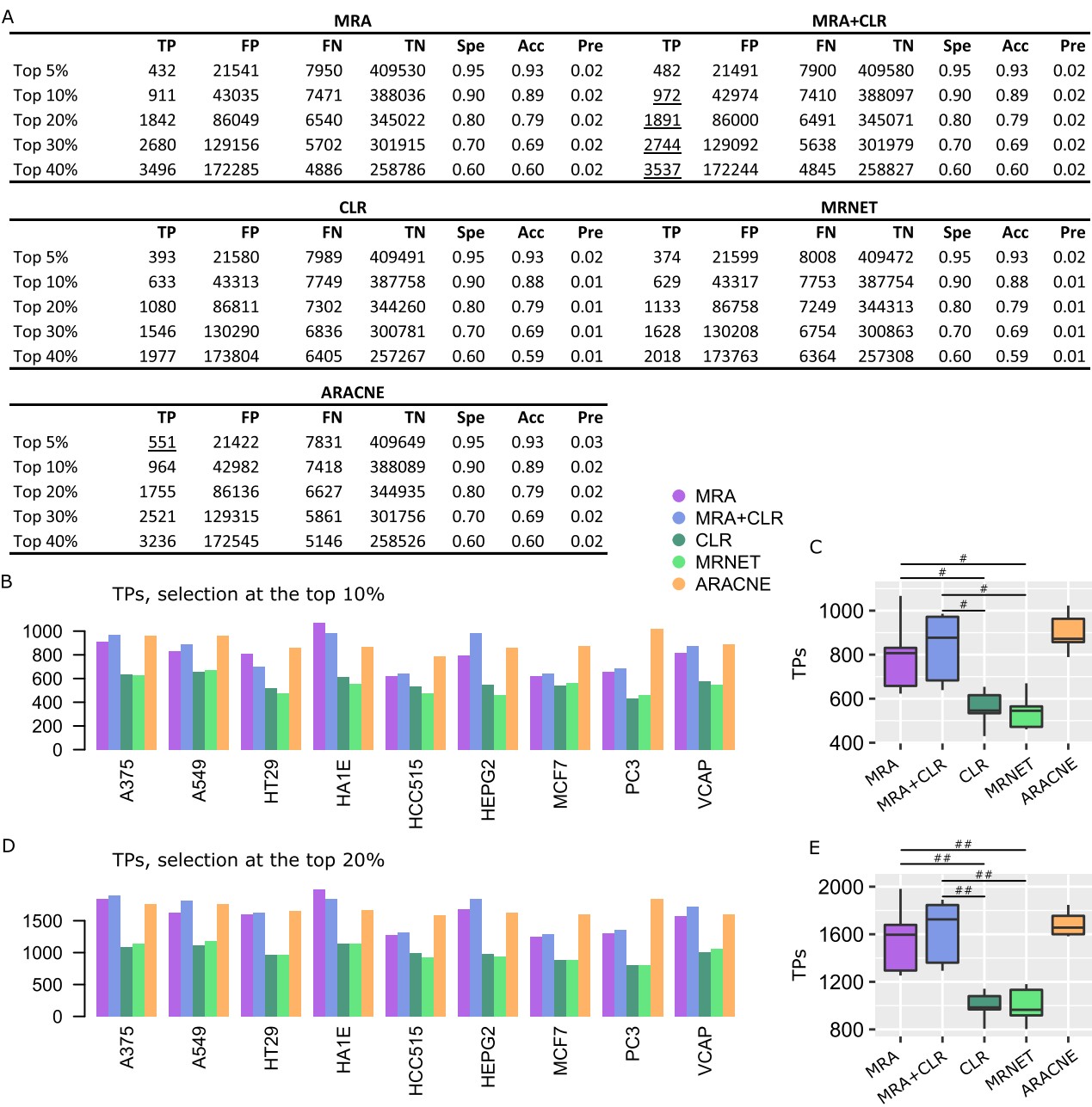

**Fig 4. Performance on L1000 shRNA data against STRING interactions (STRING score > 0.5).** (A) Representative confusion matrices for A375 cells. Spe = specificity, Acc = accuracy, and Pre = precision. Underlined values represent maxima. (B) TP numbers at the top 10% selection level. (C) Comparison between the algorithm TP numbers (Wilcoxon test, 2-sided, #P < 0.001). (D) TP numbers at the top 20% selection level. (E) Comparison between the algorithm TP numbers (Wilcoxon test, 2-sided, #P < 0.001, ##P < 0.00005).

## Discussion

We presented a particular application of MRA to large biological systems and showed its competitive performance compared to first-in-class MI-based inference methods. Obviously, MI-based methods have a much broader spectrum of application, as they do not need specific and systematic perturbations on the components of the biological system whose regulatory network is to be inferred. Nevertheless, when perturbation data are available, our results suggest

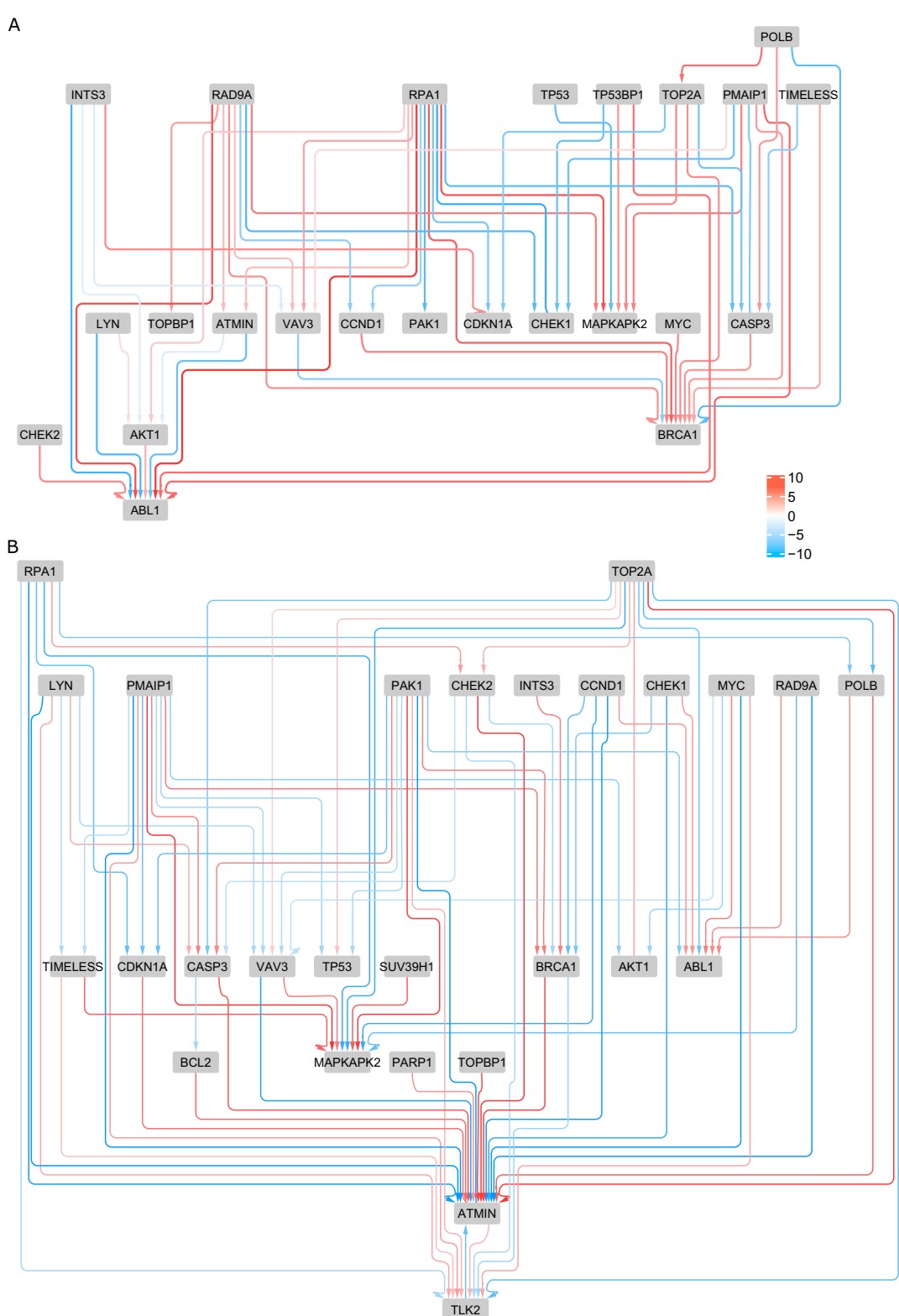

**Fig 5. Networks inferred with MRA+CLR (top 10% selection).** Genes involved in cellular response to DNA damage stimulus (GO:0006974) in (A) normal kidney cells, and (B) melanoma cells (**B**).

A

| | | MRA | | | | | | | MRA+CLR | | | | | |
|---|---|---|---|---|---|---|---|---|---|---|---|---|---|
| | TP | FP | FN | TN | Spe | Acc | Pre | TP | FP | FN | TN | Spe | Acc | Pre |
| Top 5% | 76 | 3138 | 2463 | 58584 | 0.95 | 0.91 | 0.02 | 114 | 3100 | 2425 | 58622 | 0.95 | 0.91 | 0.04 |
| Top 10% | 171 | 6256 | 2368 | 55466 | 0.90 | 0.87 | 0.03 | 246 | 6181 | 2293 | 55541 | 0.90 | 0.87 | 0.04 |
| Top 20% | 394 | 12459 | 2145 | 49263 | 0.80 | 0.77 | 0.03 | <u>482</u> | 12371 | 2057 | 49351 | 0.80 | 0.78 | 0.04 |
| Top 30% | 632 | 18647 | 1907 | 43075 | 0.70 | 0.68 | 0.03 | <u>705</u> | 18574 | 1834 | 43148 | 0.70 | 0.68 | 0.04 |
| Top 40% | 868 | 24837 | 1671 | 36885 | 0.60 | 0.59 | 0.03 | <u>943</u> | 24762 | 1596 | 36960 | 0.60 | 0.59 | 0.04 |

| | | CLR | | | | | | | MRNET | | | | | |
|---|---|---|---|---|---|---|---|---|---|---|---|---|---|
| | TP | FP | FN | TN | Spe | Acc | Pre | TP | FP | FN | TN | Spe | Acc | Pre |
| Top 5% | 131 | 3083 | 2408 | 58639 | 0.95 | 0.91 | 0.04 | <u>151</u> | 3063 | 2388 | 58659 | 0.95 | 0.92 | 0.05 |
| Top 10% | 234 | 6193 | 2305 | 55529 | 0.90 | 0.87 | 0.04 | <u>255</u> | 6172 | 2284 | 55550 | 0.90 | 0.87 | 0.04 |
| Top 20% | 404 | 12449 | 2135 | 49273 | 0.80 | 0.77 | 0.03 | 431 | 12422 | 2108 | 49300 | 0.80 | 0.77 | 0.03 |
| Top 30% | 574 | 18705 | 1965 | 43017 | 0.70 | 0.68 | 0.03 | 581 | 18698 | 1958 | 43024 | 0.70 | 0.68 | 0.03 |
| Top 40% | 727 | 24978 | 1812 | 36744 | 0.60 | 0.58 | 0.03 | 723 | 24982 | 1816 | 36740 | 0.60 | 0.58 | 0.03 |

| | | ARACNE | | | | | |
|---|---|---|---|---|---|---|
| | TP | FP | FN | TN | Spe | Acc | Pre |
| Top 5% | 123 | 3091 | 2416 | 58631 | 0.95 | 0.91 | 0.04 |
| Top 10% | 186 | 6241 | 2353 | 55481 | 0.90 | 0.87 | 0.03 |
| Top 20% | 309 | 12544 | 2230 | 49178 | 0.80 | 0.77 | 0.02 |
| Top 30% | 435 | 18844 | 2104 | 42878 | 0.69 | 0.67 | 0.02 |
| Top 40% | 542 | 25163 | 1997 | 36559 | 0.59 | 0.58 | 0.02 |

B TPs, selection at the top 10%

C

D TPs, selection at the top 20%

E

**Fig 6. Performance on L1000 CRISPR/Cas9 data against STRING interactions (STRING score > 0.5).** (A) Representative confusion matrices for A375 cells. (B) TP numbers at the top 10% selection level. (C) Comparison between the algorithm TP numbers. (D) TP numbers at the top 20% selection level. (E) Comparison between the algorithm TP numbers (Wilcoxon test, 2-sided, *P < 0.05).

that a dedicated method, relying on a modeling approach might deliver good performance in a robust fashion. The simple heuristic we proposed to prune MRA inferences, which was adapted from the CLR algorithm, provided improved performance. CLR and MRNET were systematically over performed, while ARACNE delivered variable performance. Overall, it was either clearly inferior or similar to MRA+CLR, depending on the data set. On the L1000 shRNA data, its good and less variable performance made it the algorithm of choice. ARACNE variable performance across data sets might suggest strong dependence on data characteristics such as the noise level or the dynamic range.

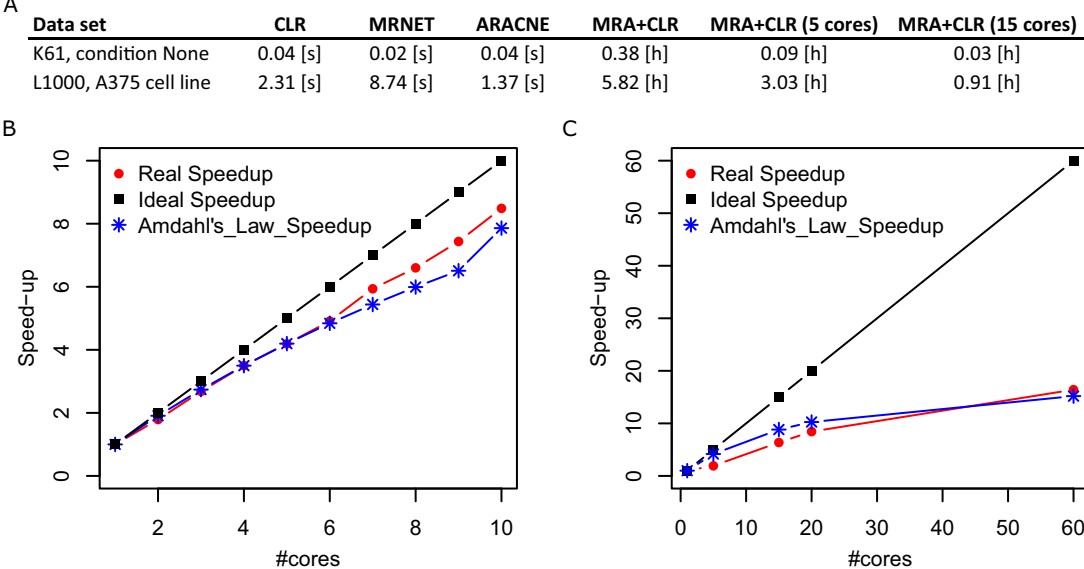

**Fig 7. Execution times.** (A) Execution times on two representative experimental networks. Note that MRA+CLR times are in hours instead of seconds. MRA times are identical to MRA+CLR times. (B) K61 data (None condition, 61×61 matrix) speedup curve. Amdahl's Law is a commonly used model for the best achievable speedup. (C) L1000 shRNA data (A375 cells, 938×938 matrix).

Execution times required by model-based algorithms such as MRA+CLR are obviously much larger than threshold- and rule-based inferences (from minutes to hours depending on the number of processors and the data set size *versus* seconds), but it remains essentially negligible compared to the time and money invested in generating experimental perturbation data.

Although the number of data sets was limited, we could notice a stronger improvement of MRA+CLR over MI-based methods with L1000 shRNA knockdown perturbation data compared to the two full KO data sets (K61 and L1000 CRISPR/Cas9). This might relate to the linearization at the heart of MRA modeling, where the error depends on the magnitude of perturbations (see our derivation of MRA through Taylor series expansion [17]). Strong perturbations such as full KOs might bring the data away from MRA area of safe application.

## Materials and methods

### Modular response analysis

We briefly recall the main MRA equations to facilitate the reading of this text, and to explain the particular way we implemented the linear algebra. We assume that the biological system is comprised of $n$ modules whose activity levels are denoted by $x \in \mathbb{R}^n$. We further admit the existence of $n$ intrinsic parameters, $p \in \mathbb{R}^n$, one per module, and each of them can be perturbed by an elementary perturbation. One can imagine $x$ reporting mRNA abundances and perturbations induced by shRNAs for instance. Lastly, we assume that there exist $S \subset \mathbb{R}^n \times \mathbb{R}^n$, an open subset, and $f:S \to \mathbb{R}^n$ of class $\mathcal{C}^1$, *i.e.*, continuously differentiable, such that

$$\dot{x} = f(x, p). \tag{1}$$

We do not need to know $f(x, p) = (f_1(x, p), \cdots, f_n(x, p))^t$ explicitly, but we need the existence of a time $T > 0$ such that all the solutions, for any $p$ and initial conditions of $x$, have

reached a steady state, *i.e.*,

$$\dot{x} = 0, \forall t > T.$$

The unperturbed, basal state of the modules is denoted $x(p^0) \in \mathbb{R}^n$ and it has corresponding parameters $p^0 \in \mathbb{R}^n$. By the application of the implicit function theorem and Taylor expansion at the first order [10,17], MRA relates the experimental observations of the global effect of perturbations to local interaction strengths, *i.e.*, the matrix $r = \left( r_{i,j} \right) = \left( \frac{\partial x_i}{\partial x_j} \frac{x_j}{x_i} \right)$ that we mentioned in Results. Such local interactions are obviously signed and non-symmetric. To compute $r$, we need to compute the relative global change induced by each elementary perturbation in each module. These values are compiled in a $n \times n$ matrix denoted $R = (R_{i,k})$ with

$$R_{i,k} = \left( \frac{\Delta x_i}{x_i} \right)_{q_k},$$

the relative difference in activity of module $i$ upon $\Delta p_k$ change induced by an elementary perturbation $q_k$ that touches module $k$ only. The relationship between observational data in $R$ and the local interactions we want to estimate in $r$ are provided by the following equations

$$\left( \frac{\Delta x_i}{x_i} \right)_{q_k} = \sum_{j \neq i} r_{i,j} \left( \frac{\Delta x_j}{x_j} \right)_{q_k}, k \neq i, \tag{2}$$

$$\left( \frac{\Delta x_i}{x_i} \right)_{q_i} = \sum_{j \neq i} r_{i,j} \left( \frac{\Delta x_j}{x_j} \right)_{q_i} + \frac{\partial x_i}{\partial p_i} (p^0) \left( \frac{\Delta p_i}{x_i} \right). \tag{3}$$

By setting $r_{i,i} = -1$, Eqs (2) and (3) can be put together in matrix form and we obtain

$$rR = -P, \tag{4}$$

where $P$ is a diagonal $n \times n$ matrix with

$$P_{i,i} = \frac{\partial x_i}{\partial p_i} (p^0) \left( \frac{\Delta p_i}{x_i} \right), i \in \{1, \cdots, n\}. \tag{5}$$

Eq (3) can be solved in two steps: $r = -PR^{-1}$ and $r_{i,i} = -1$ imply $P_{i,i}(R^{-1})_{i,i} = 1$, thus

$$P_{i,i} = \frac{1}{(R^{-1})_{i,i}}.$$

Therefore,

$$r = -[\text{diag}(R^{-1})]^{-1} R^{-1}. \tag{6}$$

In practice, relative differences in $R$ are often estimated with the more stable formula

$$R_{i,k} = 2 \left( \frac{x_i(p^0 + \Delta p_k) - x_i(p^0)}{x_i(p^0 + \Delta p_k) + x_i(p^0)} \right), \tag{7}$$

where we denote $x(p^0 + \Delta p)$ the steady-state corresponding to the changed parameters $p^0 + \Delta p$, *i. e.*, the solution of $\dot{x}(p^0 + \Delta p) = f(x(p^0 + \Delta p), p^0 + \Delta p)$.

## Parallelized and stable linear algebra

Eq (6) requires the computation of the inverse of the matrix $R$, which is less efficient and less stable than LU decomposition with pivot search [35]. These technical issues are usually irrelevant with small systems, but in applications of MRA to larger biological systems they should be addressed.

As several authors noticed, including in MRA original publication [10], the homogeneous Eq (2) is sufficient to compute $r$. Moreover, letting $i$ take the values $1, \cdots, n$, we remark that Eq (2) defines $n$ systems of linear equations of dimension $n-1$, which can be solved independently. In particular, those systems can be solved on independent processors by performing the LU decomposition with pivot search. Illustrative speedup curves are featured in Fig 7. Depending on the size of $n$, each such subsystem could itself benefit from a parallel solver if enough processors were available.

When Eq (2) is solved for each value of $i$, it is straightforward to solve Eq (3) to find $P_{i,i}$ values in case those are required:

$$\left(\frac{\Delta x_i}{x_i}\right)_{q_i} = \sum_{j \neq i} r_{i,j} \left(\frac{\Delta x_j}{x_j}\right)_{q_i} + P_{i,i} \Longleftrightarrow P_{i,i} = \sum_{j \neq i} r_{i,j} \left(\frac{\Delta x_j}{x_j}\right)_{q_i} - \left(\frac{\Delta x_i}{x_i}\right)_{q_i},$$

where Eq (4) was used for the definition of $P_{i,i}$.

## CLR, MRNET, and ARACNE computations

We used the implementation of CLR and MRNET provided by the BioConductor R package minet [28]. ARACNE was provided by the package parmigene [31]. The performance reported here reflects the performance of these specific implementations with default parameters.

## CLR heuristic adapted to MRA

We adapted the CLR normalization scheme by means of z-score computation to MRA $r$ matrix content. From $r = (r_{i,j})$ we thus derive a $Z = (z_{i,j})$ defined as follow:

$$Z_{i,\text{row}} = \frac{r_{i,j} - \frac{1}{n}\sum_{k=1}^{n} r_{i,k}}{\sigma_i}, \text{ with } \sigma_i \text{ the standard deviation of } rs\ i-\text{th row,}$$

$$z_{j,\text{col}} = \frac{r_{i,j} - \frac{1}{n}\sum_{k=1}^{n} r_{k,j}}{\sigma_j}, \text{ with } \sigma_j \text{ the standard deviation of } rs\ j-\text{th column,}$$

$$w_{i,j} = \sqrt{z_{i,\text{row}}^2 + z_{j,\text{col}}^2}, \text{ and}$$

$$Z = (\text{sign}(r_{i,j})w_{i,j}).$$

## Data sets preparation

Synthetic data were generated using the model FRANK [21] through its web server (see FRANK publication). We generated 5 networks of increasing sizes: 50TFx50TA, 100TFx100TA, 75TFx1000TA, 500TFx500TA, and 1000TFx1000TA. To obtain perturbation matrices is a built-in functionality of the online tool. The connection matrix defining the network topology is also returned by this tool.

TK61 data were obtained on multiple 96-well plates. Accordingly, we tried to stick to this format preparing data for MRA computations. We computed an $R$ matrix for each plate and

then simply averaged the relevant $R$'s for each experimental condition to obtain the averaged $R$ used in MRA. For MI-based inferences, we averaged all the relevant values.

L1000 shRNA data were extracted at level 5 (L1000 terminology) where CGSs (integration of multiple shRNA hairpins to alleviate off-target effects) were transformed into z-scores for normalization purposes by the authors of the data. Consequently, values representing the abundance of a gene were no longer positive numbers but just real numbers. Eq (7) above was adapted to compute the relative changes in MRA $R$ matrices according to

$$R_{i,k} = 2\left(\frac{\mathrm{CGS}_i(p^0 + \Delta p_k) - \mathrm{CGS}_i(p^0)}{|\mathrm{CGS}_i(p^0 + \Delta p_k)| + |\mathrm{CGS}_i(p^0)|}\right)$$

avoiding potential divisions by 0 in case of small values with opposed signs.

L1000 CRISPR/Cas9 data were averaged over replicates (also level 5).

## Performance evaluation

In the case of synthetic networks, direct access to the underlying network topology as returned by the network generator FRANK provided the reference. In the case of experimental data, we used STRING.

STRING as well as MI-based inference are devoid of direction of interaction and a sign. Therefore, the intersection of inferences with reference networks only used the upper triangular part of matrices representing the inferences (such matrices are symmetric anyway). To provide a fair comparison with MRA and MRA+CLR, we filled the upper triangular part of $r$ according to $r_{i,j} = \max\{|r_{i,j}|; |r_{j,i}|\}$, $i < j$. Moreover, as indicated in Results, STRING interactions are associated with a confidence score. Our default choice (main figures) was to only use STRING interactions with a score $> 0.5$. Supplementary tables report performance obtained using all the STRING interactions (score $> 0$), and only STRING scores $> 0.8$.

## Supporting information

**S1 Table. Confusion matrices on the synthetic data set.**
(XLSX)

**S2 Table. Confusion matrices on the K61 data set.**
(XLSX)

**S3 Table. Confusion matrices on the L1000 shRNA data set.**
(XLSX)

**S4 Table. Confusion matrices on the L1000 CRISPR/Cas9 data set.**
(XLSX)

**S1 Text. R code used to process the K61 data set.**
(ZIP)

## Author Contributions

**Conceptualization:** Meriem Mekedem, Patrice Ravel, Jacques Colinge.

**Funding acquisition:** Jacques Colinge.

**Methodology:** Meriem Mekedem, Patrice Ravel, Jacques Colinge.

**Software:** Meriem Mekedem.

**Supervision:** Patrice Ravel, Jacques Colinge.

**Validation:** Patrice Ravel.

**Writing – original draft:** Meriem Mekedem, Patrice Ravel, Jacques Colinge.

**Writing – review & editing:** Jacques Colinge.

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
