## [Decision Letter · Decision Letter 0]

13 Jan 2022

Dear Colinge,

Thank you very much for submitting your manuscript "Application of Modular Response Analysis to Medium- to Large-Size Biological Systems" for consideration at PLOS Computational Biology.

As with all papers reviewed by the journal, your manuscript was reviewed by members of the editorial board and by several independent reviewers. In light of the reviews (below this email), we would like to invite the resubmission of a significantly-revised version that takes into account the reviewers' comments.

We cannot make any decision about publication until we have seen the revised manuscript and your response to the reviewers' comments. Your revised manuscript is also likely to be sent to reviewers for further evaluation.

Sincerely,

Sarath Chandra Janga, Ph.D

Guest Editor

PLOS Computational Biology

Ilya Ioshikhes

Deputy Editor

PLOS Computational Biology

Reviewer's Responses to Questions

**Comments to the Authors:**

Reviewer #1: The authors have developed a new Modular response analysis method for biological perturbation analysis. The applications to medium and large systems show superior performances to other methods when compared to PPI. The algorithm could also parallel in multi-cores. The paper is well written and easy to follow.

Major:

1. In the case study, the authors only claim a difference between tumor and normal cells. The simple difference isn’t enough because many methods can produce a difference while the difference isn’t not necessarily meanful in biology. This is also a critical point to justify the method.

2. The method was evaluated with different proportions of predicted interactions. While the higher accuracy is valuable, it’s required to describe the evaluate the obtained values like p-values. Such values can provide a guide threshold for users to use the results.

2. It’s beneficial to show the actual running time and make comparison with other methods.

Reviewer #2: This paper proposed to use MRA to infer the protein interaction network using the data obtained through the

33 systematic perturbation of the actors. While the topic is interesting, I have a few concerns as listed below.

1. In fig 1, the green cluster seems to corresponding to FGF1, instead of WNT3A?

2. For the data there are 61genes and 11 conditions. In my understanding, there should be 11 networks estimated. Or have you inferred the networks for None-, WNT3A- and IFNg-stimulated data separately?

3. For STRING database, some of their interactions are predicted with a confidence score. Did you use any threshold in the STRING network? If so, how did you select the threshold?

4. For the performance evaluation, it is better to include other metrics like specificity, sensitivity, AUC and accuracy as well.

Without these information, it is hard for me to evaluate the quality of this work properly.

**Have the authors made all data and (if applicable) computational code underlying the findings in their manuscript fully available?**

Reviewer #1: None

Reviewer #2: Yes

PLOS authors have the option to publish the peer review history of their article (what does this mean?). If published, this will include your full peer review and any attached files.

Reviewer #1: No

Reviewer #2: No
---

## [Decision Letter · Decision Letter 1]

31 Mar 2022

Dear Colinge,

We are pleased to inform you that your manuscript 'Application of Modular Response Analysis to Medium- to Large-Size Biological Systems' has been provisionally accepted for publication in PLOS Computational Biology.

Best regards,

Sarath Chandra Janga, Ph.D

Guest Editor

PLOS Computational Biology

Ilya Ioshikhes

Deputy Editor

PLOS Computational Biology

Reviewer's Responses to Questions

**Comments to the Authors:**

Reviewer #1: I have no further comments.

Reviewer #2: The authors have full addressed all my concerns.

**Have the authors made all data and (if applicable) computational code underlying the findings in their manuscript fully available?**

Reviewer #1: None

Reviewer #2: Yes

PLOS authors have the option to publish the peer review history of their article (what does this mean?). If published, this will include your full peer review and any attached files.

Reviewer #1: No

Reviewer #2: No

---

## [Editor Report · Acceptance letter]

13 Apr 2022

PCOMPBIOL-D-21-01264R1 

Application of Modular Response Analysis to Medium- to Large-Size Biological Systems

Dear Dr Colinge,

I am pleased to inform you that your manuscript has been formally accepted for publication in PLOS Computational Biology. Your manuscript is now with our production department and you will be notified of the publication date in due course.

With kind regards,

Livia Horvath
